# The Theory of Similarity and Analysis of Dimensions for Determining the State of Operation of Structures under Difficult Loading Conditions

**DOI:** 10.3390/ma15031191

**Published:** 2022-02-04

**Authors:** Petr Louda, Vladimir Marasanov, Aleksandr Sharko, Dmitry Stepanchikov, Artem Sharko

**Affiliations:** 1Department of Material Science, Technical University of Liberec, 461 17 Liberec, Czech Republic; petr.louda@tul.cz; 2Department of Automation and Computer-Integrated Technologies, Kherson National Technical University, 73008 Kherson, Ukraine; mvsharko@gmail.com; 3Department of Transport Technology and Mechanical Engineering, Kherson Marine Academy, 73000 Kherson, Ukraine; alexandr.vladimirovich.sharko@gmail.com; 4Department of Energetics, Electrical Engineering and Physics, Kherson National Technical University, 73008 Kherson, Ukraine; dmitro_step75@ukr.net

**Keywords:** similarity theory, modeling efficiency, complex loads, residual life

## Abstract

A simulation mathematical model of the state of operability of metal structures under difficult operating conditions without stopping the equipment was developed in the form of similarity criteria found on the basis of the laws of conservation of data obtained experimentally during tensile and four-point bending tests. Criteria are proposed for the similarity of the state of the material of the samples and the products in service, in which the kinetics of destruction are determined through the rate of damage accumulation and the movement of the structural components of the material. The residual life of the equipment under conditions of complex deformation effects was determined based on the theory of similarity and the analysis of the dimensions of the parameters of acoustic emission in real time. The use of concepts and models of fracture mechanics when creating methods and criteria for assessing the results of diagnostics and monitoring allows important information about the technical state of objects to be obtained.

## 1. Introduction

The reliability and efficiency of the operation of metal structures depends on the quality of monitoring the technical conditions of the mechanical properties of materials in accordance with the requirements of international standards [1,2]. At the same time, during operation under conditions of uncertainty in the nature and magnitude of loading, the properties of materials deviate from their standard values, which requires periodic shutdowns of the equipment and diagnostic work.

According to the terminology, residual life is the duration of the safe operation of products at permissible parameters from a given moment in time to its predicted state.

The need to assess the residual life is determined if the estimated service life of the equipment in operation has been established when the terms of normative technical diagnostics are approaching, in the event of deviations in operating modes, when performing repair and restoration work.

Technical diagnostics to determine the residual resource are performed during planned shutdowns, as a rule, during major repairs. They include:External examination;Hardness measurements;Flaw detection, one of the methods of non-destructive testing: ultrasonic, magnetic particle, capillary, and acoustic emission method;Evaluation of metallographic structures;Metal cutting to control mechanical properties, chemical composition, and microstructure; andHydraulic tests.

Such a variety of applied methods and means of technical monitoring is caused by the ambiguity of the conclusions about the test results, which necessitates the introduction of their comprehensive control. However, even with the comprehensive application of non-destructive and destructive testing, there is no strictly defined sequence of their use; therefore, despite the many means used, the reliability of such estimates is low.

Interest in identifying the state of a material under load has manifested itself in the development of similar issues in related fields of science and technology. The identification of bifurcations in distributed complex networks is considered in [3]. The identification of the operability of power systems is presented in [4]. A method for calculating the parameters of mechanical tensile strength is described in [5].

Determination of the residual life of metal structures using the phenomenon of acoustic emission (AE) is presented in [6]. In [7], the AE method was used to assess the degradation of mechanical properties under complex deformation effects, in [8,9] the method was used to assess the safety factor using AE and fractal data analysis. In [10], the method was used to analyze the relationship between the energy of the low and high frequencies contained in the audio signal during the honing process.

The AE method was also used to predict fractures in composite materials [11,12], and to determine the residual life of the launching building structures of space launch vehicles [13]. The use of the similarity theory in determining the residual life is presented in [14,15,16,17].

In the above works, the deformation of the structure of materials occurs under the influence of one type of tensile or bending loading, while under real operating conditions of structures, the material is affected by loading forces, which are manifested in combined deformation conditions.

Since the magnitude of loading during the operation of metal structures is uncertain, it is not possible to fully use the existing experience under conditions of a combined loading of structures.

Theoretical estimates of the residual life of structural materials during their operation under complex deformations requires the involvement of such branches of mathematics as the theory of elasticity, theory of differential equations, theory of probability, mathematical statistics, theory of random processes, and information theory. This explains the fact that, despite the great practical importance, the issues of constructing a methodology for determining the residual resources of equipment remain unresolved.

Deviation from these states reduces the safety of operation and the resource, which makes the task of monitoring, diagnosing, and predicting the strength and deformation parameters of the operated products in real time especially crucial.

The issue of increasing the efficiency of identification processes for the state of metal structures under conditions of uncertainty in the nature of loading requires the development of new methods and technologies for non-destructive determination of the mechanical properties of materials.

## 2. Methodology

The physical essence of the described toolkit for determining the performance of products during their operation is based on the accumulation of damage and the resulting changes in the structure of materials that initiate the generation of information signals, including AE. Analysis of the theory of dimensions allows the amount of experimental material to be reduced without losing information about the current state of the structure.

Questions related to the interpretation of the data obtained in the experiment on specimens with fixed, strictly defined types of deformation, and their transfer to other more complex types of deformations under the operating conditions of products under load are related to one fundamental position. Since the experiment was performed on a less complex object, it is of interest to know what information from this experiment is contained in other more complex objects. To do this, it is necessary to clarify the fundamental similarities and differences between the functioning of the samples and real structures, and between specific conditions for changing deformations in the experiment and the operation of a metal structure.

When conducting experiments, due to the large number of multidimensional arguments, it is not always possible to find their dependencies and relationships. In this case, methods of similarity theory using complexes may be of great help. Similarity theory is the scientific basis for experimental studies of complex phenomena, using modeling methods, and methods of analogies.

The general task of similarity theory is to develop a methodology aimed at ordering information about objects that exist outside of our consciousness, and that interact both with each other and with the external environment. Methods of mathematical modeling assume the similarity of the studied phenomena and processes, while the model itself is built on the basis of the similarity derived from the analysis of conservation laws.

In practice, analogies are used as varieties of similarity, and modeling methods are used as the implementation of analogies. If all the variables describing this process are known, then homogeneous equations may be drawn up in terms of dimensions. If the equations cannot be drawn up, then it becomes necessary to apply the method of dimensional analysis.

Depending on the relative completeness of the available information, three cases of similarity and simulations are possible.

The case of full simulation and similarity. Mathematically, this case is characterized by the ratio of the parameters of the model *X* and the original *Y* [18].
(1)$$X_{j}=m_{j}Y_{j}$$
where *m_j_* is a scale factor that determines spatial coordinates and operating time.

The case of partial modeling and similarity providing similarity in either time or space. The mathematical representation of the system parameters is
(2)$$Y_{j}=\varphi \left(y_{1},y_{1},\ldots ,y_{k-j},t\right)$$
or
(3)$$Y_{j}=\varphi \left(y_{1},y_{1},\ldots ,y_{k-j},l_{x},l_{y},l_{z}\right)$$
where *l_x_*, *l_y_*, and *l_z_* are the size of local changes in the structure, and *t* is time.

The case of approximate modeling, where an error is deliberately allowed, which requires its assessment when analyzing the final results. This includes modern knowledge about the qualitative similarity and patterns of ongoing processes, their differences, and features.

A simulation model of the state of health of structures may be presented in the following form.
(4)$$\frac{y}{y_{0}}=f\left(\frac{x_{1}}{x_{10}},\frac{x_{2}}{x_{20}},\ldots ,\pi_{1},\pi_{2},\ldots \right)$$
where *y*/*y*_0_ is the desired variable, expressed in relative form, *x_i_*/*x_i_*_0_ are independent variables, and *π_i_* are complexes.

The solution is generalized, since it includes countless cases corresponding to different parameter values *x*_10_, *x*_20_, …, *y*_0_.

The mathematical description of the functional dependence of physical quantities does not depend on the choice of units of measurement, and all terms of the equation must have the same dimension. To do this, they may be converted to a dimensionless form by dividing both sides of the equation by a constant value having the same dimension. The solutions are presented in the form of complexes that combine various parameters in combinations that are determined by the very mechanism of the process. The transition from individual quantities to correctly constructed complexes that acquire the meaning of new variables the number of arguments to be reduced. The specific conditions for the functioning of systems, which were previously expressed through changes in their parameters, are now expressed through complexes.

Two physical processes are considered similar if they are subject to the same laws, and all quantities characterizing one process may be obtained by multiplying homogeneous quantities by constant numbers, i.e., similarity constants characterizing another process.

The complex reflects the relative measure of the intensity of the effects determined by the operators *D_i_* and *D_k_*.

Comparing pairwise operators in the form of relations, we obtain relative operators.
(5)$$d_{ik}=\frac{D_{i}}{D_{k}}$$

If the initial physical quantities allow equations to be composed that determine the process under study, then the complexes may be found directly from these equations.

The typical form of such equations is
(6)$$D_{1}+D_{2}+\ldots +D_{i}+D_{k}+\ldots +D_{r}=0$$
where *D* indicates differential operators, each of which determines one of the characteristic features of the effects of changes in the structure of materials under loading.

Although operators cannot be used directly to determine the properties of processes, they serve as the basis for finding the laws of constructing complexes.

The dimension of any derived quantity may be expressed as *S* = *F^α^L^β^T^γ^* where *α*, *β*, *γ* and matter 0, 1, 2 … The form of the equations describing the phenomenon does not depend on the choice of the system of units. 

Dimensional analysis may be defined as a method for studying the relationships between the numerical values of quantities that are most significant for the process under study. Its application requires knowledge of the main primary and secondary variables that affect the final result. Primary quantities are measured directly, and secondary quantities are measured indirectly through determinative equations. The number of primary and secondary quantities for such phenomena is the same.

For most problems related to mechanical systems, three basic dimensions of length *L*, time *T*, and mass *M* are used.

On the basis of the axiom of the theory of dimensions, the numerical value of a physical quantity *A* is equal to the ratio of this quantity to the unit of its measurements. Parameters of the process of changing the structure of the material under load *a*_1_, …, *a_n_*, are expressed in base units *A*_1_, *A*_2_, …, *A_n_*. The rest of the derived units *B* are established on the basis of physical laws
(7)$$B=A_{1}^{n_{1}},A_{2}^{n_{2}},\ldots ,A_{k}^{n_{k}}$$

Since all process parameters are interrelated, their subsequent values depend on the previous ones, i.e.,
(8)$$a_{n}=f\left(a_{1},a_{2},\ldots ,a_{n-1}\right)$$

The units of measurement for the first *k* quantities are set independently of each other, and the units of measurement for the remaining *n*–*k* quantities are derived.
(9)$$A_{k-1}=\overset{k}{\underset{i=1}{\prod }}A_{i}^{m_{i}},\ldots ,A^{n}=\overset{k}{\underset{i=1}{\prod }}A_{i}^{p_{i}}$$

Then, increasing the base units *A*_1_, *A*_2_, …, *A_n_* respectively by *a*_1_, *a*_2_, …, *a_k_* times, the resulting ratio may be presented in the form of complexes or similarity criteria π*_i_*
(10)$$\pi_{n-k}=f\left(1,1,\ldots ,1,\pi_{1},\ldots ,\pi_{n-k-1}\right)$$
or
(11)$$\pi_{n-k}=F\left(\pi_{1},\ldots ,\pi_{n-k-1}\right)$$
where
(12)$$\pi_{1}=\frac{a_{k+1}}{\prod_{i=1}^{k}a_{i}^{m_{i}}}=\frac{A_{k+1}}{\prod_{i=1}^{k}A_{i}^{m_{i}}},\ldots ,\pi_{n-k}=\frac{a_{n}}{\prod_{i=1}^{k}a^{p_{i}}}=\frac{A_{n}}{\prod_{i=1}^{k}A_{i}^{p_{i}}}$$

If *n* − *k* = 1, then π*_n_*_−*k*_ = const.

The complex is a generalized variable.

According to the fundamental theorem of dimensional analysis, if any equation is homogeneous with respect to dimensions, then it may be transformed to a relation containing a set of dimensionless combinations, and find dimensionless similarity criteria characteristic of this process.

The order of finding the similarity criteria consists in a reasonable choice of the fundamental variables of the process under study with the subsequent recording of the functional relationship with respect to one of the fundamental variables. Typically, this is the variable for which the experiment was performed.

The methodology for finding similarity criteria consists of a set of the following operations:Choice of fundamental variables;Records of the functional state of the process;Choice of a system of primary variables through which all the fundamental variables may be expressed;Drawing up of dimensional formulas for the fundamental variables;Records of the uniformity condition for the interaction of all indicators of the functional ratio; andDrawing up of dimensionless complexes.

It should be noted here that if it is not possible to obtain a system of dimensionless combinations, then this is a sure sign of an error, i.e., it is necessary to search for which of the function variables is missing.

In the case of multiparameter loading, to determine the safety factor, one should take into account how the deformation parameters change when the loading parameters change with respect to the limiting state. The boundary curve of the area of operability of structures under difficult loading conditions, constructed from the results obtained from samples of St3sp steel [19], is shown in Figure 1.

The condition for operability is the failure of the loading trajectory to go beyond the area of determining the state of operability of the structure. The intersection of the boundary curve with the coordinate axes is determined by the values of the strength properties of the material *σ*_B_ and *σ*_0.2_ [19].

The boundary curve equation has the form
(13)$${\left(\frac{\sigma_{b}}{\sigma_{0.2}}\right)}^{2}+\frac{\sigma_{t}}{\sigma_{B}}=1$$
where *σ**_t_* and *σ**_b_* are the results of monitoring mechanical stress during the bending and tensile tests.

For an arbitrary point A, geometrically, the residual resource coefficient *λ* is equal to the ratio of segment OB to segment OA.
(14)$$\lambda =\frac{OB}{OA}\geq 1$$

Analytically, the residual resource at point A is expressed by the equation
(15)$$\lambda =\frac{\sigma_{0.2}}{\sigma_{bA}}\left[-\frac{1}{2}\frac{\sigma_{0.2}\sigma_{tA}}{\sigma_{B}\sigma_{bA}}+\sqrt{\frac{1}{4}{\left(\frac{\sigma_{0.2}\sigma_{tA}}{\sigma_{B}\sigma_{bA}}\right)}^{2}+1}\right]$$

## 3. Materials and Methods

St3sp grade structural carbon steel was chosen as a material for the research. This steel is used for the manufacture of load-bearing structural elements. It is made from I-beam, structural channel, rolling, metal corners, cold-rolled coils, and steel sheets. ST3sp steel is well welded by gas and electric arc welding. This steel is not prone to temper brittleness. It has good technological properties.

International analogues of St3sp grade steel represented in Table 1.

Mechanical properties of steel of St3sp grade represented in Table 2.

The methods of the theory of similarity and analysis of dimensions were used as the research methods.

When solving the general problem of linking the results obtained from the samples with a prediction of the bearing capacity of structures, along with the use of well-known similarity criteria, there is a need to develop new criteria closer to the physical essence of the studied phenomena.

To determine the actual state of structural materials, information about the fracture kinetics, characterized by the rate of damage accumulation, are taken into account in strength calculations.

Since the kinetics of fracture are determined by the rate of damage accumulation, it is necessary to consider similarity criteria, in which the rate of changes in the structural components of the material, which determine the mechanical properties and performance of structures under difficult loading conditions, is included as a parameter.

The main criterion establishing the relationship between the rate of development of the phenomenon *υ*_0_, its scale *l* and time *t*, is the criterion of homochronism *π_H_*_0_.
(16)$$\pi_{H_{0}}=\frac{\upsilon_{0}t}{l}$$

The expediency of using this criterion in assessing the performance of structures lies in the transfer of the basic laws obtained on the samples to complex deformations of products under load.

The most important similarity criteria are named after the surname of famous scientists and are designated by the first letters of their surname.

The criterion for the similarity of the motion of local changes in the structure with mass *m* under the action of applied forces *F* is determined from the equation of Newton’s second law
(17)$$F=m\frac{d^{2}l}{dt^{2}}$$
and has the form
(18)$$\pi_{N_{e}}=\frac{Ft^{2}}{ml}$$

The basis for using this criterion in determining the operability of structures under loading is the provision that local changes in the structure under load move in the space of the material with their release to the surface in the form of ruptures and cracks.

The similarity of displacements of structural imperfections inside the material under stress under the action of gravity is determined by the equation
(19)$$F=m\frac{d^{2}l}{dt^{2}}=mg$$
where
(20)$$\pi_{g}=\frac{l}{gt^{2}}$$
where *g* is the acceleration of gravity.

The elastic interaction of moving structural imperfections in the material of structures under loading is described by the Cauchy criterion, for which the corresponding complex is equal to
(21)$$\pi_{k}=\frac{\upsilon_{0}}{\sqrt{\frac{E\varepsilon }{\rho }}}$$
where *E* is Young’s modulus, *ρ* is the material density, and *ε* is the relative change in linear displacement.

When a material is presented as a viscoelastic body, the similarity criteria will be the Reynolds, Froude, and Strouhal numbers.

The Reynolds number *R_e_* characterizes the speed of movement of the main accumulations of dislocations in the direction of the outer surface of the product. This number manifests itself as the ratio of inertial forces to internal friction forces.
(22)$$R_{e}=\frac{\upsilon_{0}l}{\nu }$$

The Froude number *F_r_* expresses the speed of movement of dislocations in the peripheral areas of the zone of changes in the structure of materials *υ*_1_, and manifests itself as the relative value of their gravity.
(23)$$F_{r}=\frac{\upsilon_{1}^{2}}{gl_{1}}$$
where *l*_1_—is the linear dimension from the zone of moving excitations to the outer edge of the product.

The Strouhall number *S_t_* expresses the impulsive motion of dislocations under the action of applied stresses.
(24)$$S_{t}=\frac{\upsilon_{2}t}{l}$$
where *υ*_2_ is the rate of growth of movement of dislocations, and *t* is the pulse alternation time.

Analysis of the above-mentioned criteria is extremely useful for selection of variables that determine the physical essence of the process of determining the state of health of structures under difficult loading conditions, and developing the corresponding complexes.

## 4. Building the Simulation Model

Taking into account the motion of dislocations in a material as in a viscoelastic medium in the case of unsteady motion of incompressible dislocation imperfections, the Navier–Stokes equation in vector form may be used.
(25)$$\frac{\partial {\vec{\upsilon }}_{3}}{\partial t}+\left({\vec{\upsilon }}_{3}\nabla \right){\vec{\upsilon }}_{3}-\vec{F}+\frac{1}{\rho }\nu \mathrm{grad}P-\frac{1}{\rho }\nu \nabla^{2}{\vec{\upsilon }}_{3}=0$$
where *F* is volumetric force, *ν* is kinematic viscosity, *P* is pressure, $\left({\vec{\upsilon }}_{3}\nabla \right){\vec{\upsilon }}_{3}$ is inertial force, $\frac{1}{\rho }\nu \mathrm{grad}P$ is pressure force, and $\frac{\partial {\vec{\upsilon }}_{3}}{\partial t}$ is non-stationary force.

The physical meaning of the applicability of this equation is expressed in the form of the equality to zero of the sum of pressure forces, volume force, inertial force, and internal friction.

The development of the process depends on the relationship between these forces. Therefore, for the stationary case where $\frac{\partial {\vec{\upsilon }}_{3}}{\partial t}=0$, the development of the process depends on the value of the relative operator υ→3∇υ→31ρν∇2υ→3, which determines the ratio of inertial force to internal friction force.

In the stationary case, the Navier–Stokes equation may be written as
(26)$$\left({\vec{\upsilon }}_{3}\nabla \right){\vec{\upsilon }}_{3}-\vec{F}+\frac{1}{\rho }\nu \mathrm{grad}P-\frac{1}{\rho }\nu \nabla^{2}{\vec{\upsilon }}_{3}=0$$
and contains four homogeneous operators and three complexes.

To create a deterministic simulation model for identifying the state of metal structures under conditions of uncertainty in loading, it is proposed to use discrete interaction of particles connected by elastic bonds [20].

The potential energy is a function of the displacement field *u(n)*
(27)$$\begin{array}{l} \Phi =\Phi_{0}+\underset{n}{\sum }\Phi \left(n\right)u\left(n\right)+\frac{1}{2}\underset{n,n^{\prime }}{\sum }\Phi \left(n,n^{\prime }\right)u\left(n\right)u\left(n^{\prime }\right)+\\ +\frac{1}{3!}\underset{n,n^{\prime },n^{\prime \prime }}{\sum }\Phi \left(n,n^{\prime },n^{\prime \prime }\right)u\left(n\right)u\left(n^{\prime }\right)u\left(n^{\prime \prime }\right)+\ldots \; \end{array}$$
where *Φ*_0_ is the energy of a linear chain in equilibrium, and *n′, n″ are* numbers of interacting particles.

For offsets *u(n,t),* the kinetic energy of such a chain is
(28)$$T=\frac{m}{2}\underset{n}{\sum }{\left[\frac{\partial u\left(n,t\right)}{\partial t}\right]}^{2}$$

The difference in kinetic *T* and potential *Φ* energy defines the Lagrange function *L*
(29)$$L=\frac{m}{2}\underset{n}{\sum }{\left[\frac{\partial u\left(n,t\right)}{\partial t}\right]}^{2}-\frac{1}{2}\underset{n,n^{\prime }}{\sum }\Phi \left(n,n^{\prime }\right)u\left(n,t\right)u\left(n^{\prime },t\right)+\underset{n}{\sum }q\left(n,t\right)u\left(n,t\right)$$
where *q(n,t)* is the external force.

Considering that
(30)$$\frac{d}{dt}\frac{\partial L}{\partial \dot{u}}-\frac{\partial L}{\partial u}=0$$
the equation of vibrational motion of particles takes the form
(31)$$m\ddot{u}\left(n,t\right)+\underset{n^{\prime }}{\sum }\Phi \left(n,n^{\prime }\right)u\left(n^{\prime },t\right)=q\left(n,t\right)$$

The vibrations of atoms do not initiate a wave. This requires external excitations caused by structural changes. Since all atoms vibrate in the same way, it is enough to consider only one chain, which is deformed as the excitations propagate along it.

Bias *u_n_* = *Ae^i(ωt-kxn)^* corresponds to a longitudinal wave that propagates at a speed *υ*_0_ *= ω/k*, where *k = 2π/λ* is the wave vector.

The displacement of an atom in the absence of damping is
(32)$$u_{n}=Ae^{i\left(\omega t+\phi_{n}\right)};\;u_{n-1}=Ae^{i\left(\omega t+\phi_{n-1}\right)};\;u_{n+1}=Ae^{i\left(\omega t+\phi_{n+1}\right)}$$
where *ω* is the frequency, and *φ* is the oscillation phase.

The equation of motion for an atom *n* with mass m has the form
(33)$$m\frac{\partial^{2}u_{n}}{\partial t^{2}}=\Phi \left(u_{n+1}+u_{n-1}-2u_{n}\right)$$
where *m* is the mass of an atom, and *Φ* is the force constant.

In moving nanostructured objects, there is a short-range action of discrete structures and a long-range action of an elastic continuum in which disturbances propagate.

Applying the methods of continuum mechanics, it is possible to reduce the equations of motion to combined equations with reduced parameters. The boundary conditions will be the equality to zero of the velocity of movement of dislocations on the surface of the product when a crack opens, and the act of energy emission stops ${\vec{\upsilon }}_{3}=0$ at large distances due to the attenuation of elastic waves in the material.

With this in mind
(34)$$\left\{\begin{array}{c} \nabla^{2}{\vec{\upsilon }}_{3}=\mathrm{grad}\;\mathrm{div}{\vec{\upsilon }}_{3}-\mathrm{rot}\;\mathrm{rot}{\vec{\upsilon }}_{3}\\ \left({\vec{\upsilon }}_{3}\nabla \right){\vec{\upsilon }}_{3}=\mathrm{rot}{\vec{\upsilon }}_{3}+{\vec{\upsilon }}_{3}\\ \mathrm{div}{\vec{\upsilon }}_{3}=0 \end{array}\right.$$

Assuming that the volumetric forces of structural change have the potential Π, that is, *F* = −gradΠ, the Navier–Stokes equation is reduced to the form
(35)$$\frac{\partial {\vec{\upsilon }}_{3}}{\partial t}+\mathrm{rot}{\vec{\upsilon }}_{3}+{\vec{\upsilon }}_{3}=\mathrm{grad}\left(\upsilon_{3}^{2}+\Pi +\frac{P}{\rho }\right)-\nu \mathrm{rot}{\vec{\upsilon }}_{3}$$

This equation, together with the equation $\mathrm{div}{\vec{\upsilon }}_{3}=0$, is a closed nonlinear system of four second-order partial differential equations with four unknown functions ${\vec{\upsilon }}_{3i}\left(i=1,2,3\right)$ and *P*. These equations should be supplemented by the equation of the vibrational motion of particles during the propagation of signals from structural disturbances during loading (31). The model obtained in this way will be a simulation model of the state of health of structures under difficult loading conditions.

## 5. Results and Discussion

The safety margin of metal structures during their operation is determined by the magnitude of the mechanical stress *σ*, tensile strength values *σ_B_*, and fluidity *σ*_0.2_ in bending and tensile tests, which in turn depend on the elastic moduli *E*, material density *ρ*, masses *m*, and geometric dimensions of the sample *l*, and are manifested in the form of changes in deformations under a load determined through energy *W* and strength *F*. These quantities are fundamental variables.

The dimensions of these variables are presented in Table 3.

Based on the determinative equations, it is possible to express the dimensional equation for the residual life coefficients *λ* in the form of a functional relationship of fundamental variables with their exponents
(36)$$\lambda =f\left({\left(ML^{-1}T^{-2}\right)}^{a},{\left(ML^{-1}T^{-2}\right)}^{b},{\left(ML^{-3}\right)}^{c},{\left(M\right)}^{d},{\left(L\right)}^{e},{\left(M^{0}L^{0}T^{0}\right)}^{f},{\left(ML^{2}T^{-2}\right)}^{g},{\left(MLT^{-2}\right)}^{h}\right)$$
in the form of a functional relationship of fundamental variables with their exponents
(37)$$\lambda =f\left(\frac{\sigma l^{2}}{F},\frac{El^{2}}{F},\frac{\rho l^{3}}{m},\varepsilon ,\frac{W}{Fl}\right)$$
or
(38)$$\lambda =f\left(\pi_{1},\pi_{2},\pi_{3},\pi_{4},\pi_{5}\right)$$

This equation expresses the dependence of *λ* on the dimensionless complexes *π*_i_:
$\pi_{1}=\frac{\sigma l^{2}}{F}$ characterizes the relative stresses at constant load for a sample and a metal structure;$\pi_{2}=\frac{El^{2}}{F}$ characterizes changes in mechanical properties during loading of a sample and a product during its operation;$\pi_{3}=\frac{\rho l^{3}}{m}$ characterizes the relative density of the sample material and the metal structure and determines, in the general case, the propagation velocity of stress waves;$\pi_{4}=\varepsilon$ characterizes the relative changes in deformation in the sample and in the product under a constant loading force; and$\pi_{5}=\frac{W}{Fl}$ characterizes the change in energy and disturbing force in a full-scale sample and in an exploited product.

Analysis of the considered similarity criteria and their physical interpretation show that for the processes of changes in the structure of the material under load and the occurrence of internal stresses, Newton’s criterion is the closest in its physical essence (18). If we describe the components of this criterion in terms of the corresponding dimensions, multiplying and dividing the result obtained by *L*, and combining the closest obtained representations of the variables, according to Table 3, we obtain a new similarity criterion.
(39)$$\pi_{5}=\frac{ML}{T^{2}}\frac{T^{2}}{ML}\frac{L}{L}=\frac{ML^{2}}{T^{2}}\frac{T^{2}}{MLL}=\frac{W}{Fl}$$

When applied to the assessment of the operability of structures under load, this criterion establishes a correspondence between the energy of structural change *W* and the disturbing force *F*.

Similar transformations may be performed with the Froude criterion (23). By expressing the components of this expression in terms of the corresponding dimensions, dividing, and multiplying the formula for the dimension of the Froude criterion by a constant number *MT*^2^/*L* we obtain
(40)$$F_{r}=\frac{L^{2}}{T^{2}}\frac{T^{2}}{LL}\frac{L}{MT^{2}}\frac{MT^{2}}{L}$$

Taking into account the fact that in the resulting expression *MT*^−2^*L*^−1^ = *σ* and *MLT*^−2^ = *F* we obtain a new criterion *π*_1_
(41)$$\pi_{1}=\frac{\sigma l^{2}}{F}$$

When applied to the assessment of the operability of structures, it expresses the relative changes in stresses under a constant load.

Since, when assessing the performance of structures, the main process under study is the change in internal stresses and the associated change in the mechanical properties of the material, the Cauchy criterion is the most acceptable similarity criterion. By squaring Formula (20), expressing the components of the resulting expression in terms of the corresponding dimensions, and uniting the closest variables, we obtain a new similarity criterion π_2_
(42)$$\pi_{2}=\frac{\upsilon^{2}\rho }{E\varepsilon }=\frac{L^{2}}{T^{2}}\frac{M}{L^{3}}\frac{LT^{2}}{M}=\frac{L^{2}MLT^{2}}{T^{2}LLLM}=\frac{E}{F}l^{2}$$

By writing out the formula for the displacements of structural imperfections of the material (21) in terms of the dimensions of the quantities included in it, and multiplying and dividing the result by *ML*^−1^, considering that *ML*^−3^ = *ρ*, we obtain one more similarity criterion *π*_3_
(43)$$\pi_{3}=\frac{L^{2}T^{2}}{T^{2}LL}\frac{M}{L}\frac{L}{M}=\frac{\rho l^{3}}{M}$$

This criterion makes it possible to describe in more detail the change in the properties of materials of products through stresses in the structure that responsible for the generation of emerging information signals.

When studying the elastic deformations of structures under the influence of external forces, Poisson’s ratio is used as the main criterion
(44)$$\mu =\left|\frac{\varepsilon_{1}}{\varepsilon_{2}}\right|$$
where *ε*_1_ = Δ*l*/*l*, is the relative longitudinal deformation, and *ε*_2_ = Δ*d*/*d* is the relative lateral deformation.

A criterion based on these definitions $\pi_{4}=\varepsilon$ will be another complex that determines the state of health of structures during their operation.

Complexes *π_i_* allow the determination of how many times any parameter of the model will change when others change.

The choice of a system of dimensionless criteria is predetermined by the ability to concentrate all the uncertainty associated with the experiment into one of the dimensionless complexes. One of the most promising means of monitoring the state of critical facilities during their operation, taking into account the operating time of equipment in modes of extreme and peak loads on the structure material, is the use of a complex based on the implementation of the effect of acoustic emission, which allows the initial stage of changes in the structure of materials to be determined. In this case, it becomes possible to diagnose on continuously operating equipment.

The main requirement for the experiment is the equality of the values of the root mean square errors in measurements on samples and real structures. In this work, this condition is met due to a single recording unit with digital indication of AE signals for tensile and bending tests (Figure 2).

Samples for four-point bending tests were cut from sheet metal with a size of 300 × 20 × 4 mm, for tensile testing with a size of 223 × 37 × 3 mm. Identification of the structural features of the bending deformation mechanisms was carried out during the deformation of the samples on an installation operating on the principle of a given deformation. The tests were carried out with registration of the load and the corresponding deformation with simultaneous fixation of the moments of occurrence of AE signals in compliance with ASNT, ASTM, and ISO9001 standards.

The measuring setup used broadband sensors with a bandwidth of 0.2…0.5 MHz. The information-measuring system used in the experiment provided indication, registration, and pre-processing of AE signals with its further storage in the computer memory using a storage oscilloscope (RIGOL DS1052E Digital oscilloscope). The results of AE measurements are shown in Figure 3 and Figure 4.

In the loading diagram, the moments of occurrence of AE signals are noted not only by characteristic transitions at the stages of elastic, plastic, and pre-fracture loading, but also by a more detailed structure change within these stages (Table 4).

The AE method allows real-time recording of the processes of violation of the structure of metals in the dynamics of their changes. The fixation of deformation transitions by AE methods shows that these changes occur earlier than it follows from tensile tests. This makes it possible to establish the degree to which a material is approaching the state of destruction.

To implement the method for determining the state of operability of structures under difficult loading conditions, it is necessary to perform measurements on samples under loading and certain types of deformation, such as bending and tension, which is reflected in the complex π_4_ to assess the state of operability of metal structures. Such measurements were performed in [21,22] and presented in the form of their digital processing in determining the information parameters recommended for technical diagnostics.

An effective way to study the dynamic processes of changing the structure of materials is an active experiment. The purpose of this experiment is to establish correlations between mechanical, deformation, and information diagnostic parameters of AE under various test conditions.

Studies have shown that peak amplitudes, amplitude-frequency distributions, amplitude-time distributions, energy spectrum, and density of AE signals are recommended as information parameters of AE signals for use in bending. For stretching, parameters such as the total number of pulses and the intensity of AE acts are also added to them.

The agreement with the similarity criteria proposed in this work made it possible to select from the informational parameters of the AE that most clearly reflect the essence of the loading process. It was found that in relation to the AE phenomenon, the closest criterion is *π_3_* = *ρl/m*, with the only difference in interpretation being that the density of the AE signals is now defined as the number of pulses in the investigated range. Then the number of experiments using the AE phenomenon in diagnosing the performance of products is significantly reduced.

The density of AE signals was chosen as an information parameter in acoustic measurements. For ST3sp steel, this parameter bends by an order of magnitude when biological objects are deformed.

The experimental values of the fundamental variables to determine the state of health of structures under difficult loading conditions are presented in Table 5.

The density and energy of the AE signal reflect changes in the mechanical properties of the materials.

The density of the AE signal is determined by the relationship
(45)$$N=\frac{N_{\sum }}{t_{H}}$$
where *N*_Σ_ is the number of crossings of the AE signal threshold level.

AE signal energy
(46)$$W=\overset{n}{\underset{i=i_{c}}{\sum }}\left|u_{i}-u_{base}\right|$$
where *i_c_* is the sample number corresponding to the time of arrival of the AE signal, *u_base_* is the displacement of the AE signal relative to the zero level, and *u_i_* is the current offset of the AE signal.

They are an indicator of structural transformations during loading of metal structures.

The presented AE information was processed together with directly measured mechanical characteristics, which reflect the strength properties of the material.

The calculated values of the similarity criteria are presented in Table 6.

To calculate complexes π_2_ and π_3_, data on the geometric dimensions of the samples were used, as well as reference data on the density *ρ* (7850 kg/m^3^) and Young’s modulus *E* (200 GPa).

The exploited product and the sample are described by the same dimensionless equations
(47)$$\lambda_{1}=f\left(\pi_{1},\pi_{2},\pi_{3},\pi_{4},\pi_{5}\right)$$
(48)$$\lambda_{2}=f\left({\pi_{1}}^{\prime },{\pi_{2}}^{\prime },{\pi_{3}}^{\prime },{\pi_{4}}^{\prime },{\pi_{5}}^{\prime }\right)$$

The only difference is in dimensionless complexes.

The values of the complexes presented in Table 4 reflect different moments of changes in the structure of the materials. In this case, the primary event was a change in load.

To assess the state of operability of the operated structures at a given time, it is necessary not only to determine the mechanical properties, but also to determine the properties that are sensitive to local structural changes manifested in microplastic deformation and changes in local micro-stresses. The strength properties decrease not only with an increase in the service life, but also with an increase in the strength conditions. Accumulation of damage under complex dynamic deformation effects characterizes structural degradation processes such as deformation and destruction of cementite in pearlite columns, evolution of dislocation substructure, and formation of carbide precipitates in the bulk of ferrite grains. This leads to a significant decrease in the strength properties of metals, and the residual resource in general.

The reason for the degradation of the mechanical properties of the metal is strain aging, which is explained by the fact that during long-term operation there is a change in the microstructure of the metal, a decrease in strength, embrittlement, and a decrease in the bonding forces between crystal grains.

The mechanisms of degradation of the mechanical properties of metals are:Accumulation of dislocations at barriers;Evolution of dislocation structures;Formation and growth of nuclei of new carbide phases;Penetration of atoms into tetrahedral voids of the ferrite lattice; andDecomposition of cementite and fragmentation of pearlite grains.

When assessing the performance of metal structures using AE methods, it is necessary to solve the inverse problem by the values of the AE signals at the moments of recording the occurrence of the AE phenomenon, to determine the initiating values of the loads on the product. This may be performed using approximating polynomials.

The approximating polynomials for the dependences of mechanical stress on the density of AE signals *σ*_p_(*N*) or *σ*_и_(*N*) for stretching and bending are
(49)$$\sigma_{t}\left(N\right)=-4\cdot 10^{4}N^{4}+6.54\cdot 10^{4}N^{3}-3.61\cdot 10^{4}N^{2}+7313N+0.86$$
(50)$$\sigma_{b}\left(N\right)=\frac{-74.51N^{2}+200.3N+0.43}{N^{2}-3.63N+3.82}$$

The approximating polynomials for the dependences of the relative deformation on the density of the AE signals *ε_t_*(*N*) or *ε_b_*(*N*) for stretching and bending are
(51)$$\varepsilon_{t}\left(N\right)=\frac{-0.094N^{2}+0.348N-0.108}{N^{2}-3.394N+2.929}$$
(52)$$\varepsilon_{b}\left(N\right)=\frac{0.238N^{2}+0.022N+0.002}{N^{2}-0.436N+0.049}$$

When approximated, the resulting smoothing curve does not necessarily pass through all the nodal points and is a general trend and pattern. Application of approximating polynomials to determine σ*_t_*, σ*_b_*, ε*_t_*, and ε*_b_* allows for a transformation from the analysis of the input information obtained on the samples to the input information obtained on the product.

To identify the state of metal structures, it is necessary to transform the graph of Figure 1 in the form of a plane *λ = f*(*N_t_, N_b_*).

The change in the residual life coefficient under combined loading conditions is presented in the form of a response surface to the disturbing effect of the accumulation of damage to the structure of materials. When constructing the response surface, we used calculations according to Formula (15), performed for the values σ_B_ and σ_0.2_ for a given steel grade, and changing values *σ_t_* and *σ_b_*. In this case, the presented surface is the domain of existence of function (15), where the values of strength characteristics included in it are expressed through approximation polynomials. In addition to a visual representation, such a visualized form of changes in the coefficient λ has its practical meaning. After performing acoustic measurements and the necessary operations to determine the density of acoustic signals, it is possible to identify the value of λ as the third coordinate of the intersection *N_t_, N_b_* with this surface. The healthy state area is characterized by the values λ ≥ 1. In Figure 5, this area is above the plane λ = 1.

The quantitative values of the results of the AE measurements performed on a single-girder bridge crane during its loading are presented as point A of the load curve 2 (Figure 3), which shows the current values *N_t_*,*N_b_* corresponding to the applied load.

When processing the results of the AE measurements, the values of the density and energy of the AE signals were determined using Formulas (45) and (46).

The results of the acoustic measurements and processing of the obtained information for the construction of complexes of the exploited product are presented in Table 7.

By values *N*, presented in Table 5, and using Formulas (50)–(53), it is possible to find the values *σ* and *ε* corresponding to the applied loads *F*.

To identify the load applied to the product in use, it is necessary to identify from the whole variety of characteristics in Table 3 and complexes in Table 4, only those in which the density of AE signals corresponds to point A during tension and bending (Table 7). The results of this identification are presented in Table 8.

This part of the research represents the part of the experiment that may be carried over from the assessment of the state of the sample to the state of the product in use.

The calculated values of the dimensionless complexes of the exploited product are presented in Table 9.

Considering $\left[\pi_{1},\pi_{2},\pi_{3},\pi_{4},\pi_{5}\right]$ as a vector, in accordance with (47.48) with known values of the complexes for tension and bending of the samples and the operated product, it is possible to find the normalized scalar product
(53)$$R_{\lambda_{1}\lambda_{2}}=\frac{\sum_{i}\pi_{i}{\pi_{i}}^{\prime }}{\sqrt{\sum_{i}\pi_{i}^{2}}\sqrt{\sum_{i}\pi \prime_{i}^{2}}}$$

With the same characteristics of the devices used in the study of the residual life coefficient on the sample and the operating product, we have
(54)$$\lambda_{2}=R_{\lambda_{1}\lambda_{2}}\lambda_{1}$$

Substitution of the data in Table 6 and Table 7 into Formula (53) gives the following value of the correlation coefficient $R_{\lambda_{1}\lambda_{2}}=0.94$, to assess the values of the residual life coefficients on the sample λ_1_ and used product λ_2_, calculated by Formulas (15) and (54), we obtain λ_1_ = 1.38, and λ_2_ = 1.30.

It is not the residual resource ratio that is of practical interest. λ_2_, and the value of the safety factor Δλ, determine the excess of this value over the limit value λ = 1 determining the performance of the structure
(55)$$\Delta \lambda =\lambda_{2}-1$$

Determination of the safety factor of a controlled item under complex dynamic loading using AE signal density as a recorded parameter without stopping the equipment for diagnostics is visualized in Figure 6.

The values calculated based on Formula (15) λ_2_ using approximation polynomials (50–53) and found based on the proposed complexes π_i_’ differ on average by (15 ± 3)%. This allows us to conclude about the adequacy of the proposed estimates for determining the safety factor, and the possibility of applying the theory of similarity and dimensional analysis to obtain information on determining the operability of structures under difficult loading conditions without stopping the equipment for diagnostics.

## 6. Conclusions

The use of similarity criteria and dimensional analysis when constructing complexes of a simulation model for determining the state of operability of structures in difficult loading conditions was aimed at ordering information about objects that exist outside our consciousness, and interact with each other. The developed method for determining the operability of structures under difficult loading conditions makes it possible to concentrate the estimation uncertainty into one of the dimensionless complexes. It was proven that one of the most promising means of monitoring the state of critical objects during their operation, taking into account the operation time of equipment in modes of extreme and peak loads on the structure material, is the use of a complex based on the implementation of the effect of acoustic emission, which allows the initial stage of changes in the structure of materials to be determined. The obtained results may serve as a model representation for predicting the residual life and safety factor of metal structures.The found similarity criteria make it possible to recalculate the results obtained from experiments on samples subjected to tension and bending to determine the residual life of equipment operated under difficult loading conditions. This may be used as a basis for the formulation of recommendations for reducing operational loads for continuously operating equipment and the turnaround time for servicing metal structures.The main advantage of the practical application of the theory of similarity and analysis of dimensions is that it may be applied to complex phenomena, knowing only the set of variables that determine the essence of the phenomenon under study without losing information about the current state of the structure. The use of material constants in the formulas of complexes testifies to the versatility of the methods for constructing complexes. When switching to other loading, it is enough just to change the numerical values of these constants.The use of dimensional analysis provides three main benefits:
The number of variables under consideration is always reduced;New dimensionless variables make it possible to conduct experiments more economically and efficiently; andRatios derived from dimensionless variables are general and not limited to any particular system of units.


## Figures and Tables

**Figure 1 materials-15-01191-f001:**
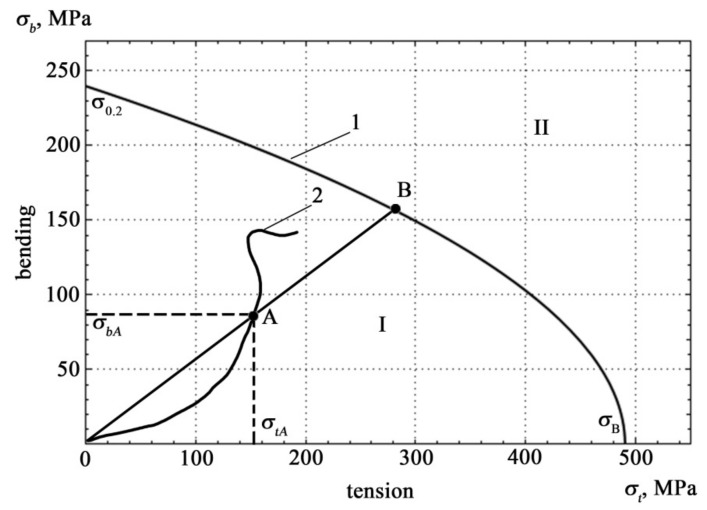
Boundary curve 1 and loading trajectory 2 in the coordinates of mechanical stress (I—area of operability, II—area of destruction) [19].

**Figure 2 materials-15-01191-f002:**
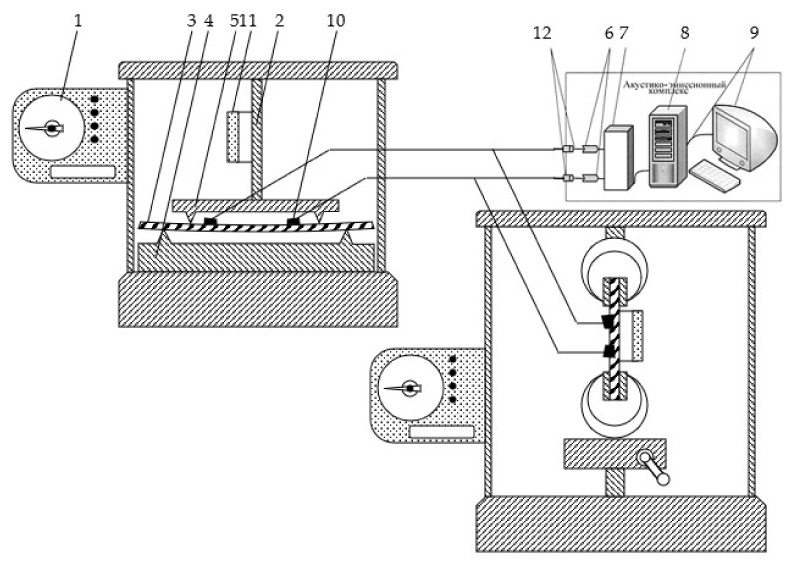
Diagram of an experimental setup for AE measurements during tensile and four-point bending tests: 1. Force measuring mechanism; 2. Deformation mechanism; 3. Controlled sample; 4. Support; 5. Indenter; 6. Filter unit; 7. Analog-to-digital converter; 8. Block for accumulation and processing of information; 9. Recorder; 10. Piezoelectric sensor; 11. Tensometer; and 12. Blocks of preliminary amplification.

**Figure 3 materials-15-01191-f003:**
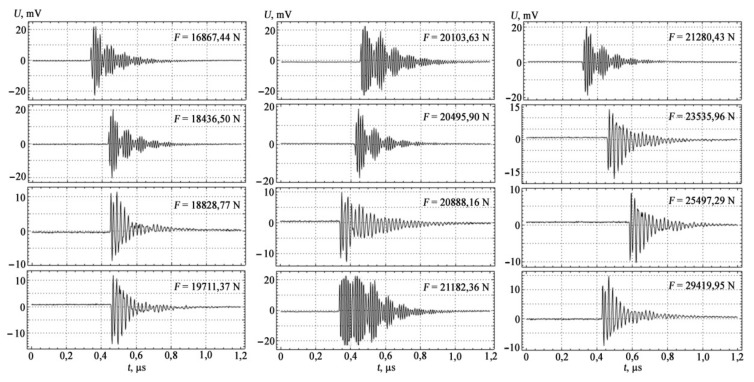
Structure of the spectra of AE signals for tension.

**Figure 4 materials-15-01191-f004:**
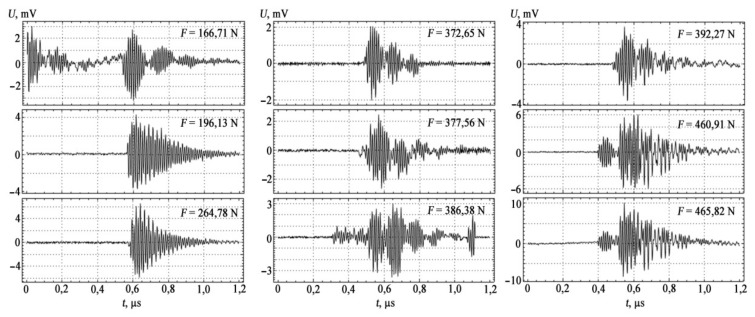
Structure of the spectra of AE signals for bending.

**Figure 5 materials-15-01191-f005:**
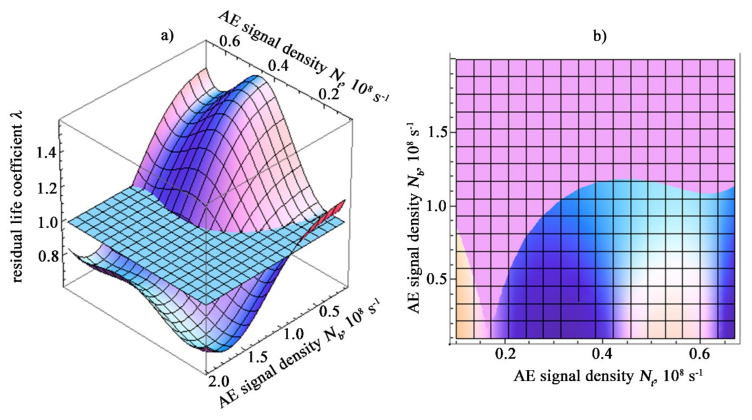
Graphical interpretation of the residual life of specimens made of ST3sp steel under difficult loading conditions: (**a**) The surface of the safety factor in the coordinates of the density of the AE signals (the horizontal plane corresponds to the value *λ* = 1 and separates the operability region from the fracture region). (**b**) An orthogonal projection showing the line of intersection of the safety factor surface with plane *λ* = 1.

**Figure 6 materials-15-01191-f006:**
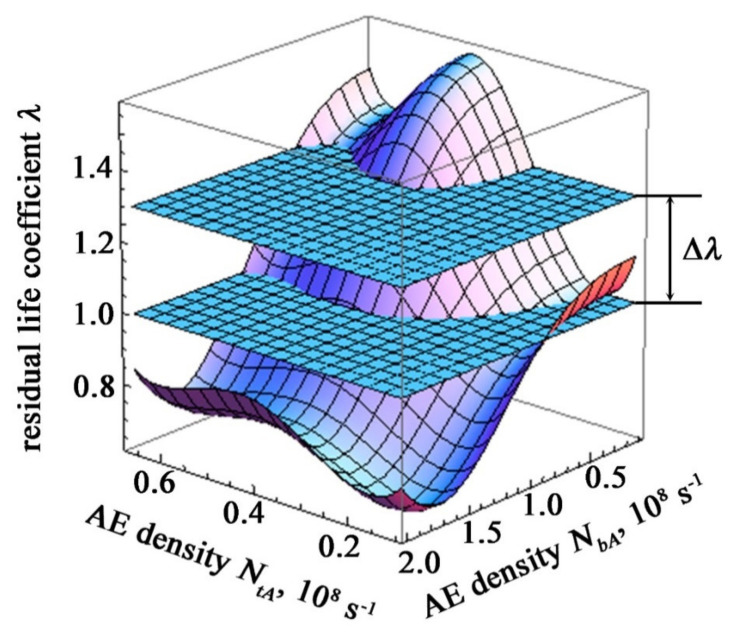
Determination of the safety factor and the state of operability of structures under difficult loading conditions.

**Table 1 materials-15-01191-t001:** International analogues steel of St3sp grade.

Germany	USA	Japan	France	Belgium	China
1.0038 St37-3	A284GrD M1017	SS330 SS400	E24-2 S234JRG2	FE360BFN FED1FF	Q235A Q235B

**Table 2 materials-15-01191-t002:** Mechanical properties of steel of St3sp grade.

Standard	*σ*_B_, MPa	*σ*_0.2_, MPa	*δ*, %
ISO 4995-78	370–490	206–245	23–26

**Table 3 materials-15-01191-t003:** Dimensions of variables that determine the state of health of metal structures.

Variable	Designation	Functional Relationships
Speed	*u*	*LT* ^−1^
Time	*T*	*T*
Mechanical stress	*σ*	*ML* ^−1^ *T* ^−2^
Deformation	*ε*	*M* ^0^ *L* ^0^ *T* ^0^
Young’s modulus	*E*	*ML* ^−1^ *T* ^−2^
Weight	*m*	*M*
Length	*l*	*L*
Force	*F*	*MLT* ^−2^
Energy	*W*	*ML* ^2^ *T* ^−2^
Density	*ρ*	*ML* ^−3^

**Table 4 materials-15-01191-t004:** Amplitude parameters of AE signals under tension and four-point bending of samples of St3sp steel.

Work Hardening Zones	Loading Force *F*, H	Maximum Amplitude *U*_max_, mV	Average Amplitude *U_cp_*, mV
Tension			
I zone of elastic deformation	16,867.44	3	0.1
	18,436.50	20	0.3
	18,828.77	11	0.2
	19,711.37	12	0.2
II zone of plastic deformation	20,103.63	22	7.0
	20,495.90	19	3.5
	20,888.16	9.8	2.2
III zone of yield and pre-fracture	21,182.36	22	1.4
	21,280.43	21	0.4
	23,535.96	14	2.8
	25,497.29	7.6	0.1
	29,419.95	15	0.3
Bend			
I zone of elastic deformation	166.71	3	0.03
	196.13	4.3	0.01
	264.78	6.6	0.01
II zone of plastic deformation	372.65	2.1	0.007
	377.56	2.5	0.008
	386.38	3.0	0.029
	392.27	3.7	0.047
III zone of yield and pre-fracture	460.91	6.1	0.184
	465.82	10.0	0.473

**Table 5 materials-15-01191-t005:** Experimental values of fundamental variables in tension and four-point bending of full-scale samples of St3sp steel.

Loading Force *F*, (N)	AE Signal Density *N*, 10^8^ s^−1^	AE Energy *W*, 10^−3^ rel. Units (1)	Mechanical Stress *σ*, (MPa)	Elongation (Deflection) *l*, (mm)	Relative Deformation ε, (%)
Tension					
16,867.44	0.47	0.302	281.1	0.375	0.17
18,436.50	0.40	4.955	307.3	0.463	0.20
18,828.77	0.19	1.616	313.8	0.477	0.21
19,711.37	0.16	2.855	328.5	0.577	0.26
20,103.63	0.55	16.000	335.1	0.591	0.27
20,495.90	0.41	4.354	341.6	0.705	0.32
20,888.16	0.33	2.360	348.1	0.737	0.33
21,182.36	0.65	34.000	353.0	1.923	0.86
21,280.43	0.44	5.464	354.7	4.189	1.88
23,535.96	0.29	4.307	392.3	11.022	4.94
25,497.29	0.25	1.804	424.9	13.330	5.98
29,419.95	0.22	2.403	490.3	17.390	7.80
Bend					
166.71	1.39	0.302	93.8	6.511	1.13
196.13	1.02	0.512	110.4	6.815	1.19
264.78	1.37	0.859	149.0	8.011	1.39
372.65	1.92	0.083	209.7	14.011	2.44
377.56	1.88	0.158	212.4	14.312	2.49
386.38	1.86	0.376	217.4	14.823	2.59
392.27	1.54	0.256	218.7	15.418	2.69
460.91	1.66	1.270	234.5	23.901	4.17
465.82	1.73	2.323	235.3	24.030	4.19

**Table 6 materials-15-01191-t006:** Calculated values of similarity criteria for samples specimens of St3sp steel.

Tension					
Loading force *F*, (N)	π_1_, 10^−3^	π_2_	π_3_, 10^−6^	π_4_, 10^−3^	π_5_, 10^−5^
16,867.44	2.34	1.67	4.43	1.70	4.77
18,436.50	3.57	2.32	8.34	2.00	58.04
18,828.77	3.79	2.42	9.12	2.10	17.99
19,711.37	5.55	3.38	16.14	2.60	25.10
20,103.63	5.82	3.47	17.34	2.70	134.70
20,495.90	8.28	4.85	29.45	3.20	30.13
20,888.16	9.05	5.20	33.64	3.30	15.33
21,182.36	61.62	34.91	597.57	8.60	83.47
21,280.43	292.48	164.92	6177.09	18.80	6.13
23,535.96	2024.92	1032.33	112,521.00	49.40	1.66
25,497.29	2961.10	1393.79	19,904.10	59.80	0.53
29,419.95	5039.87	2055.83	44,192.80	78.00	0.47
Bend					
Loading force *F*, (N)	π_1_	π_2_	π_3_	π_4_, 10^−3^	π_5_, 10^−4^
166.71	23.85	50,858.50	0.023	11.30	2.78
196.13	26.14	47,360.70	0.027	11.90	3.83
264.78	36.11	48,475.10	0.043	13.90	4.05
372.65	110.47	105,358.00	0.231	24.40	0.15
377.56	115.23	108,504.00	0.246	24.90	0.29
386.38	123.63	113,733.00	0.274	25.90	0.66
392.27	132.53	121,200.00	0.308	26.90	0.42
460.91	290.64	247,883	1.147	41.70	1.15
465.82	291.68	247,924.00	1.166	41.90	2.07

**Table 7 materials-15-01191-t007:** Experimental values of input information for calculating dimensionless complexes of an exploited product.

Loading Force *F*, (N)	AE Signal Density *N*, 10^8^ s^−1^	AE Energy *W*, 10^−3^ rel. Units (1)	Mechanical Stress *σ*, (MPa)	Elongation (Deflection) *l*, (mm)	Relative Deformation ε, (%)
Tension					
40,848.80 ± 1225.46	0.02	4.64	147.12 ± 4.41	0.50	0.06
Bend					
1579.41 ± 47.38	0.90	1.16	90.62 ± 2.72	1.61	0.18

**Table 8 materials-15-01191-t008:** Dimensionless complexes of samples made of ST3sp steel under loading.

Loading Force *F*, (N)	π_1_	π_2_	π_3_	π_4_	π_5_
Tension					
16,867.44 ± 506.02	2.34 × 10^−3^	1.67	4.43 × 10^−6^	1.70 × 10^−3^	4.77 × 10^−5^
Bend					
264.78 ± 7.94	36.11	4.85 × 10^4^	4.30 × 10^−2^	13.90 × 10^−3^	4.05 × 10^−4^

**Table 9 materials-15-01191-t009:** Dimensionless complexes of the exploited product.

Loading Force *F*, (N)	π_1_	π_2_	π_3_	π_4_	π_5_
Tension					
40,848.80 ± 1225.46	2.15 × 10^−1^	155.78	6.40 × 10^−6^	6.64 × 10^−4^	1.01 × 10^−3^
Bend					
1579.41 ± 47.38	14.75	3.28 × 10^4^	1.48 × 10^−4^	1.89 × 10^−3^	4.57 × 10^−5^

## Data Availability

The data presented in this study are available on request from the corresponding author.

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
