# Peer review of "The Theory of Similarity and Analysis of Dimensions for Determining the State of Operation of Structures under Difficult Loading Conditions"

_materials, 2022, doi:10.3390/ma15031191_

Round 1

Reviewer 1 Report

Authors of the manuscript deal with the possibility of introducing similarity criteria in order to cope with multidimensional loading of metallic materials, i.e. to account for different type of loadings that have consequence on the remaining life of the structure. The study comprises of several different parts and theories, all trying to be incorporated in a single similarity model.

Although the manuscript has some valid points, major improvements are needed to merit recommendation for “Materials” journal.

  1. Section “Relative works” should be joined to “Introduction”, because introduction generally gives the overview of previous references.
  2. There is a large number of self-referencing which is not entirely necessary. Also, there is a large number of regionally oriented references.
  3. Reference for Eq. (1) is missing, or it is original work of authors?
  4. No properties are given for the studied material ST3sp.
  5. No details about tensile and 4-point bending tests are given, e.g. standards used, images and dimensions of specimens, resulting stress-strain diagrams…

Author Response

Thanks for the careful consideration of our manuscript and the necessary comments, the elimination of which made it possible to significantly improve the work.

1. The relative works section has been merged with the Introduction section. Part of the text is excluded, part of the text is moved to the methodology section

2. Carefully reviewed all the necessary links. Excluded irrelevant, regionally oriented references, reduced self-citations

3. Added reference to the fundamental work [18] (Handbook on Experimental Mechanics: Edited by Albert S. Kobayashi. – Wiley & Sons, Incorporated, John, 1992. – 1074 p.) after equation (1).

4. Reference information has been added to the text of the article in the Materials and Methods section, shaking the description of the properties of the investigated material ST3sp.

5. Expanded Results and Discussion section. The details of the experiment are given. Figure 3. The structure of the spectra of AE signals for different loading is excluded as not informative. Instead, Figure 3. The structure of the spectra of AE signals for tension and Figure 4. The structure of the spectra of AE signals for bending have been added with explanations in the text. References are made to the used ASNT, ASTN, ISO standards. Recorded acoustic signals are presented synchronized with the stress-strain diagram.

Reviewer 2 Report

The following points need to be addressed before the publication of this manuscript.

  1. What is the relevance of these scientific studies to the study under your work:

“Modeling and 80 identification of interactions of biological objects” [10] and “Management of 81 transformations of the tourist attractiveness of regions into strategic management” [11].

  1. What is the significance of “Figure 3. The structure of the spectra of AE signals for different loading” in this particular study. I couldn’t find any useful information which could be supported by Figure 3.

  1. Moreover, since this paper deals significantly with the theory and mathematical modeling, theory about “Damping Factor” and “Settling time” must be considered to explain the AE signal in Figure 3.

Author Response

We express our gratitude for the careful review of our work The Theory of Similarity and Analysis of Dimensions for De-termining the State of Operation of Structures Under Difficult Loading Conditions P.Louda, V.Marasanov, A.Sharko, D.Stepanchikov and A.Sharko and comments made, the elimination of which significantly improved the quality of work.

  1. All necessary links have been carefully reviewed. Excluded irrelevant, regionally oriented references, reduced self-citations
  2. Expanded Results and Discussion section. The details of the experiment are given. Figure 3. The structure of the spectra of AE signals for different loading is excluded as not informative. Instead, Figure 3. The structure of the spectra of AE signals for tension and Figure 4. The structure of the spectra of AE signals for bending have been added with explanations in the text. References are made to the used ASNT, ASTN, ISO standards. Recorded acoustic signals are presented synchronized with the stress-strain diagram.

Round 2

Reviewer 1 Report

Thank you for acknowledging the comments.